# Antimicrobial Use by Peri-Urban Poultry Smallholders of Kajiado and Machakos Counties in Kenya

**DOI:** 10.3390/antibiotics12050905

**Published:** 2023-05-13

**Authors:** Florence Mutua, Gideon Kiarie, Miriam Mbatha, Joshua Onono, Sofia Boqvist, Emily Kilonzi, Lawrence Mugisha, Arshnee Moodley, Susanna Sternberg-Lewerin

**Affiliations:** 1International Livestock Research Institute, Animal and Human Health Program, P.O. Box 30709, Nairobi 00100, Kenya; f.mutua@cgiar.org (F.M.); g.kiarie@cgiar.org (G.K.); e.kilonzi@cgiar.org (E.K.); a.moodley@cgiar.org (A.M.); 2Department of Public Health, Pharmacology and Toxicology, Faculty of Veterinary Medicine, University of Nairobi, P.O. Box 30197, Nairobi 00100, Kenya; miriammbatha@students.uonbi.ac.ke (M.M.); joshua.orungo@uonbi.ac.ke (J.O.); 3Department of Biomedical Sciences and Veterinary Public Health, Swedish University of Agricultural Sciences, SE-750 07 Uppsala, Sweden; sofia.boqvist@slu.se; 4Department of Wildlife and Animal Resources Management, College of Veterinary Medicine, Animal Resources and Biosecurity, Makerere University, Kampala P.O. Box 7062, Uganda; lmugisha@covab.mak.ac.ug; 5Ecohealth Research Group, Conservation & Ecosystem Health Alliance, Kampala P.O. Box 34153, Uganda; 6Department of Veterinary and Animal Sciences, University of Copenhagen, Stigbøjlen 4, 1870 Frederiksberg C, Denmark

**Keywords:** antimicrobial resistance, veterinary drug, animal health, food security, value chain

## Abstract

Antimicrobial use (AMU) is a major driver of antimicrobial resistance (AMR). An understanding of current practices can lead to better targeting of AMU-reducing interventions. An analysis of the distribution and current usage of veterinary drugs in peri-urban smallholder poultry systems in Kenya was undertaken. A survey among poultry farmers and key informant interviews with agrovet operators and other players in the value chain was conducted in Machakos and Kajiado counties. Interview data were analyzed using descriptive and thematic approaches. A total of 100 farmers were interviewed. The majority (58%) were > 50 years old, and all kept chickens, while 66% kept other livestock. Antibiotics constituted 43% of the drugs reportedly used on the farms (*n* = 706). These were mostly administered by the farmers themselves (86%) through water (98%). Leftover drugs were stored for later use (89%) or disposed of (11%). Incineration was the main method for the disposal of leftover drugs and empty containers. As described by the key informants (*n* = 17), the drug distribution chain relied on agrovet shops that were supplied by local distributors and pharmaceutical companies, which, in turn, supplied drugs to the farmers. Farmers reportedly purchased drugs without prescriptions and rarely observed the withdrawal periods. Drug quality was a concern, especially for products requiring reconstitution.

## 1. Introduction

In low- and middle-income countries (LMICs), the poultry sector has a significant potential to alleviate poverty, enhance food and nutrition security, and promote gender equality [1,2]. Animal disease is an important constraint in this sector. Antibiotics are vital in the management of bacterial diseases [3]; however, their use selects for antimicrobial resistance (AMR) [4,5,6] and hence overuse and misuse should be avoided. In many LMICs, antimicrobial use (AMU) is rarely monitored [7] or regulated, which is a challenge to the control of AMR [8]. However, reports about the emergency of AMR in LMICs have been recently published. At the global level, AMU in animals has been increasing, from an estimated 63,151 (±1650) tons in 2010, to about 131,109 tons in 2013 [9,10], and is further expected to increase by 2030 [11].

Poultry production is increasing in Kenya. In 2006, there were about 37.3 million birds of which 84% were indigenous breeds, 8.4% were layers, 5.7% were broilers, and 1.8% included other poultry species [12]. Ensuring good health is important for production. Although biosecurity is an important disease control measure, its implementation in smallholder farms is challenging [13] and, as such, is rarely observed in many farms within the region [14]. Antimicrobials are frequently used to promote flock health and to enhance production. Agrovets are shops authorized by the Veterinary Medicines Directorate (VMD) to sell agricultural products under the management of qualified veterinary professionals [15]. The Veterinary Medicines Directorate is the agency that regulates dispensing of veterinary medicines in Kenya. Farmers often source poultry products from local agrovets without veterinary prescriptions and may administer the drugs themselves [16,17]. A few studies have reported AMR in bacteria in the poultry value chain [18,19]. Moreover, there is no national monitoring of AMU and AMR in Kenya that can be used to identify and target interventions or evaluate impacts where interventions have been implemented.

AMU in animals can impact public health. Failure to observe drug withdrawal periods can yield animal products containing antibiotic residues [20]. These can have direct, adverse effects on consumers (allergic reactions in sensitive people) as well as contribute to AMR [21]. AMR occurs when a drug cannot prevent the growth of or kill a microorganism which was previously sensitive to it (i.e., drugs become ineffective against microorganisms for which they were previously useful). Bacteria, in the presence of antibiotics, experience a selection pressure to enhance their survival, by acquiring and/or expressing antibiotic resistance genes (ARGs) and disseminating mobile ARGs among bacterial populations [22]. Resistant bacteria can be transferred from food animals to humans through direct contact with the animals, or indirectly through consumption of contaminated animal-source foods or contact with contaminated environments [5,23,24]. Some of the ARGs identified in food bacteria have also been identified in humans, providing indirect evidence for transfer via food [23]. The impact of AMR in animals is yet to be quantified, but in 2019 a global estimate of 4.95 (3.62–6.57) million human deaths associated with bacterial AMR was predicted [25].

To address AMR and prioritize action areas, surveillance data (AMU and AMR) are indispensable. In addition, although farmer extension services are an option for addressing animal health challenges [26], they are often hindered by inadequate coverage of animal health delivery systems and poor infrastructure [27]. To inform the design of a context-relevant intervention to mitigate AMR, a baseline study was designed to understand drug use practices in selected smallholder poultry systems in Kenya, through interviews with farmers and other actors in the value chain.

## 2. Results

### 2.1. Characteristics of Study Participants

A detailed description of the farm demographics is given by Mbatha et al. (in preparation). Briefly, a total of 100 farmers were interviewed. The majority (58%) were >50 years old and all kept chicken (100%), with some also keeping ducks (17%) and turkeys (16%). In terms of the chicken numbers, 36% had ≤ 100 birds, 45% had >100–500, and 21% had over 500 birds. Two-thirds (66%) also owned other livestock species besides poultry, such as sheep, cattle, goats, donkeys, and pigs. Almost all farms (96%) had at least one person owning a mobile smartphone. Interviews were conducted with 17 key informants: 12 agrovet outlets, 1 main distributor, 2 county veterinary officials, and 2 experts who had worked in the pharmaceutical sector and were at the time of the study working on animal feeds.

### 2.2. Drug Usage in Humans and Non-Poultry Species

Farmers owning other livestock species (*n* = 66) were asked to indicate how drug use in those other animals differed from that in poultry in terms of the amounts used and their frequencies of use. For these other species, compared to use in poultry, drugs were reportedly used more frequently (57% of the respondents) and in higher quantities (75%).

Fourteen (21%) of the sixty-six farmers reported using the same drugs to treat different livestock species. Actual sharing of the same products across species was reported by 11 of these 14 farmers; the remainder bought the same drug but used it for specific animals. In addition to treating sick animals, veterinary drugs were also given for disease prevention (78%), to boost growth (41%), and to treat human infections (8%). Farmers were asked to state if they sometimes had used drugs intended for human treatment in animals; nine farmers (with flock sizes of 25–1310) did so, for the following reasons (Table 1):

### 2.3. Drug Usage in the Farms

A total of 721 AMU practice records were noted, but actual use by the farmers—as determined from responses to the question on when they had used the product—was 706 records (i.e., 15 records were reported by farmers who had previously not used the product and were therefore not included in the analysis). The 706 records included antibiotics (304; 43%), acaricides (85; 12%), dewormers (72; 10%), disinfectants (62; 8%), anticoccidials (12; 2%), vitamins (152; 21%), and other products (20; 2%), which included one observation for aloe vera and one for toxin binder. The distribution of the recorded drugs used in the two study sites is given in Figure 1.

A relatively higher number of responses on antibiotic use was reported in Machakos (46%; 154/325) than in Kajiado (40%; 150/361). The antibiotic products most frequently reported (*n* = 304) included Tylodoxy (15%), Aliseryl (12%), Biosol (8%), Oxysol (7%), and Egocin (6%) (Figure 2). A high percentage of the products reported were classified as having multiple ingredients (65%, *n* = 196) (Table 2). Tetracycline (65%), macrolides (39%), sulfonamides (31%), polymyxins (18%), aminoglycosides (18%), and quinolones (1%) were among the drug categories observed in the study.

At the time of the study visit, farmers were asked to specify the last time they had used specific products in their farms. For antibiotics (*n* = 304), farmers reported using the drugs in the previous 1 year (223; 73%), 1 month (46; 15%), 2 weeks (8; 3%), 1 week (7; 2%), and in the same week the research team visited the farm (20; 7%). The reported routes of administration (*n* = 303) for the antibiotics were water (98%), feed (1%), and topical applications, especially for eye infections (1%). The products (*n* = 303) were administered by the farmers themselves (self-administration) (86%), agrovet sellers (7%), private veterinarians (7%), and other animal health providers (1%), as well as friends of the poultry farmers (4%) (Figure 3). Self-administration (95%) and help from agrovets (7%) were reported more in Machakos than in Kajiado, while administration by animal health providers (9%) and friends (6%) was more common in Kajiado than in Machakos (Table 3). Farmers used veterinary products for a variety of reasons, including boosting growth, treatment of diseases, and to manage stress (Table 4). Leftover antibiotics were reported in 18% of the antibiotic records (56/304); they were either stored for later use (89%; *n* = 56) or disposed of (11%).

### 2.4. Veterinary Product Distribution

The veterinary product distribution chain included the following actors: farmers, agrovets, distributors, and pharmaceutical companies (Figure 4). Farmers mainly sourced products from agrovet outlets. However, large farms may receive products directly from the pharmaceutical companies if they meet criteria specified by the supplier (quantity ordered, etc.). The “Pull-push” concept was reported, where the technical team generates the order (demand) and the sales department pushes the product.

For visits, the company representative must keep a record of all products sold and the appointments made. They receive a commission that is reportedly pegged on performance, including meeting sales targets and introducing new customers. The team use motorcycles and can also carry some products to take to the farmers when they visit. The direct farm suppliers do not consider whether the farm has a veterinarian or not, as long as the farm has met the criteria to be supplied with the products. Agrovet outlets either receive products from the pharmaceutical companies, directly, or from the distributors. Some agrovets have a credit arrangement with the distributors and will not receive products from pharmaceutical companies. It was mentioned that the distributors also provided agrovets with a basket option of different commodities (unlike the pharmaceutical companies, which were said to be limited to selected products). Distributors are supplied by the pharmaceutical companies. The working arrangement included agreeing on monthly targets, based on the sales potential of the distributor. The pharmaceutical company may pledge to give technical support or assist with the marketing through branding (shops, vehicles). Pharmaceutical companies also conduct market intelligence. This applies in the case where one company has run out of a particular product, and another company, as a response, either lowers the price of the product to increase the sales or gives the product to its distributors to pump it into the market (so that by the time the first company’s product comes back to the market, there is no space for their product). New products are introduced if they are known to have good profit margins, or if suggested by the sales team to fill a specific gap in the market. It costs about USD 4000 to register a new product on the Kenyan market, as reported by one participant. Another way that farms reportedly receive antibiotics is through hatcheries, which deliver these along with day-old chicks, exposing the birds to antibiotics very early in life. Hatcheries also advise farmers regarding the vaccination schedules to follow.

### 2.5. Access and Use of Veterinary Products

The usual practice is that a farmer visits an agrovet, mentions the clinical signs they have observed in their flock, and asks for treatment advice. Symptoms that were frequently reported by farmers included loss of appetite, snoring, and diarrhea. It was mentioned that there are times when a farmer just walks into an agrovet shop and says they want a poultry drug. The agrovet attendant will ask what the problem is, and a conversation often leads to the farmer receiving advice from the agrovet seller, based on what he or she perceives to be the problem. One participant observed that advice given at the agrovet shop will depend on the knowledge of the store attendant. It was reported that some farmers are mindful of the price of the product and will consider this when purchasing products. In one agrovet shop, it was mentioned that some farmers refer to Poltricin—a product containing oxytetracycline and vitamins—as “dawa ya mia” meaning a drug that does not cost more than KES 100 and will specify this when they visit the agrovet.

There are times when farmers visit agrovet shops with specific products in mind. They may present empty sachets, bottles, or old labels, especially if they have used the drug before and obtained good results. It was mentioned that some farmers present product names written on a paper while others show pictures of the products they specifically wish to purchase. In addition, there are those that come with prescriptions from veterinarians. While the agrovet seller may influence some farmers to take a different product if the specific product they have requested is not stocked by the agrovet, others are difficult to convince. One participant noted “… Sometimes farmers insist on getting what they used in the past if it helped”. Farmers also educate each other on what to use, based on past experiences. Farmers who are not conversant with poultry drugs were said to be keen when they visited and would follow the directions given by the agrovet seller. A farmer may also falsely say that their animal health provider recommended the products they visited the agrovet to buy.

Although some farmers visit the agrovet shops to buy drugs for disease prevention, it was mentioned they are usually advised against that practice by the attendants.

Tylosin was said to be “a fast mover”, especially during the cold season when respiratory problems are common. Tylodoxy (tylosin and doxycycline) and ESB3 (sulphonamides) were also said to be popular. Coccidiostats, dewormers, vaccines (especially for Newcastle and gumboro disease), and supplements are among the products frequently sold by agrovets, in addition to respiratory disease products. For disease prevention, farmers are advised to vaccinate, and in the case of reduced growth, vitamins are recommended. “Broiler booster”, which is non-antibiotic-containing, is reportedly sold to farmers who complain of ”stunted growth”. However, there are farmers who insist on obtaining Poltricin to promote growth, especially in day-old chicks. The use of Egosin for prophylaxis was also reported and was said to contain vitamins that enhance growth.

As many poultry products are administered by farmers, agrovet sellers cannot be sure if the product sold was administered as advised in the agrovet shop or not. In response to this, one agrovet reported writing directions for use on the drug package for the farmers to follow. In some agrovet shops, farmers may be asked to bring a bird for observation before the attendant gives a recommendation. One agrovet visited farms once or twice a week, during which they took samples for analysis or conducted necropsies. Another agrovet said they ensured all their customers obtain receipts, which have the telephone contacts which the farmer can call in case they forget the instructions given. Agrovets also provide vaccination schedules for farmers ordering day-old chicks. Another concern is that of farmers not understanding the prescription given by experts (for example, 5 mg per 10 L of drinking water). They lack the means to determine or measure the amount of drug and the required volume of water. An example was given for the product Biotrim, where farmers are asked to put a certain number of spoonfuls in five liters of water, raising concern over what size of spoon should be used.

Farmers were also reported to consult with untrained people, a practice that was thought to occur in the community where, for example, a farmer rather than a trained animal health professional teaches other farmers about poultry diseases and proposes the drug to be used, meaning that only difficult cases are referred to animal health professionals. It was reported that some farmers see no problem with making purchases from agrovet outlets, as this is what they usually do for vaccines. Farmers would stop using the antibiotics when they see that the birds have recovered or improved. The treatment regimen is thus stopped before the recommended dosage is completed. Cases of farmers keeping stocks of veterinary drugs on their farms were also reported, perhaps for use whenever the need arose.

There were concerns over farmers instructing farm workers over the phone on what to do when they called to report cases of sick birds. There was also a perceived risk of farmers buying drugs without consulting, of underdosing, or even of using the wrong medicines. The possibility of workers sharing the products with the neighbors or selling these to earn extra income was also mentioned. It was noted that workers may prefer to slaughter and sell sick birds if the condition is deemed severe. Farmers rarely observe drug withdrawal periods. A case of a farmer producing 10 trays of eggs in a day was given, to illustrate the income concern. The farmers would want to know what to do with the eggs produced during the withdrawal period (especially with a ready market available). Instances where drug withdrawal periods are not indicated on the sachets were observed. A suggestion to advise farmers to use products with short withdrawal periods was made. For regulators, difficulty in proving that a farmer has failed to observe drug withdrawal periods was reported, a concern that could complicate the reporting of non-compliance. Examples of reported misinformation included putting meat in the fridge to reduce drug concentration levels, and incubating eggs instead of selling them. Although the focus was on poultry, it was mentioned that dairy farmers would argue that bulking of milk dilutes the drug concentration.

### 2.6. Quality of Veterinary Products and their Disposal

Quality was perceived to be questionable, especially for bulky products that must be constituted and put into smaller units (in which case the concentration is less than in the original). An example was given where 20 kg of tetracycline is packed into smaller doses, and sold using wrongly printed labels by untrustworthy people. This was also reported for nutritional supplements which were said to contain unknown substances, and as a consequence poultry failed to perform as indicated on the package. Agrovets are required to sell products that are approved for sale; however, there was a complaint that attendants lack knowledge of how to identify counterfeit products. Some feed dealers, although not registered to sell drugs, reportedly also keep a few antibiotics. Cases where untrustworthy individuals buy cheap products but, unethically, attach the label for a genuine product of a company that is known to sell well, were reported. This was thought to be the reason why some companies regularly change their labels and use barcodes that are not easy to tamper with to circumvent the problem (it was noted that this measure would only apply to those companies that can afford it).

Expiry was said to be a problem for injectable products, which are rarely kept by farmers. Sachets with small quantities of such drugs are not a major concern as they are unlikely to remain long enough to expire. Nonetheless, it was reported that farmers are advised to discard the products after five days. In the case of agrovets, they return any expired product to the distributor (who normally will pick it up on their next visit to the shop). Agrovets can also access the services of companies that dispose of the products (an estimated fee of KES 400 (USD 3) per kilogram was mentioned). Agrovets have adopted different measures to avoid having expired products, including ordering small quantities at a time and confirming expiry dates when receiving products; retaining new stock in cartons (this ensures that stock in the older boxes that are nearing expiry are sold first); avoiding products that come with short expiry periods; if products have a short shelf-life then only those that sell fast are chosen; and checking product expiry dates on a regular basis (e.g., when cleaning the shelves) and prioritizing selling products that are about to expire.

The farmers were asked to specify how they disposed of drug containers and packages after their use on the farms. Burning was the most frequently reported method of disposal in both study sites (60% and 70% in Machakos and Kajiado, respectively) (Figure 5). A higher proportion of farms in Machakos disposed of their waste by burying (8/50; 16%) compared to farms in Kajiado (2/50; 4%).

Similar data were obtained from the key informants (Table 5). A participant noted that it is not uncommon to find drug packages kept on raised surfaces within the poultry houses.

### 2.7. Farmers’ Need for, and Willingness to Share, Information

The farmers were informed of the plan to develop an Information and Communication Technology (ICT) tool where they could share information on disease occurrence, drug usage, and disease control, and also receive feedback from veterinarians. The information needs highlighted by the farmers are summarized in Figure 6 below. A detailed description is given in Table 6.

Almost all farmers (98%) were willing to share disease information with agrovet outlets and veterinarians registered on an ICT platform.

## 3. Discussion

AMR has serious public health implications and addressing inappropriate AMU in animals is an important part of combating AMR. To improve AMU, knowledge about the drug distribution chain is needed. This study investigated antibiotic use, its drivers, and its distribution chain in two counties in Kenya, a country where poultry production is an important source of food and income for over 60% of households [12,28]. Inadequate feed and diseases are, however, important challenges faced by poultry farmers [29], and in this study many diseases, such as infectious coryza, Newcastle disease, coccidiosis, and helminthiasis were reported, similar to other studies in Kenya [30,31]. Poultry farmers also kept other species of livestock (66% in the current study). It has been argued that poor households prefer to diversify into keeping more than one species of livestock to take advantage from each, and to spread the risks associated with the farming systems [32]. Farmers reported using poultry drugs to treat infections in other livestock species. The frequency of drug use was higher in these other species than in poultry. Interventions to reduce AMR should consider the complex nature of the value chain and its interaction with other connected systems.

Antimicrobials were frequently used to ensure better flock health and to enhance production, indicating a lack of understanding of responsible AMU. The drugs were often administered by the poultry farmers themselves, with a few farmers consulting agrovets and other animal health providers. Administration of veterinary drugs by farmers themselves is a common practice in the region [16,17]. Furthermore, veterinary drugs were also used for the purpose of prophylaxis within farms and in a few instances to treat human infections. Similarly, the use of human medicines to treat sick animals was reported. Antibiotics were the drugs most frequently used by farmers in both counties (43%). Most products were found to contain multiple classes of antibiotics. For example, Tylodoxy contains tylosin (macrolide) and doxycycline, a tetracycline while Aliseryl contains erythromycin (macrolide), streptomycin (aminoglycoside), oxytetracycline (tetracycline), colistin (polymyxin) and vitamins. A previous study in agrovet outlets in Nairobi found tetracyclines and sulfonamides to be amongst the most commonly purchased antibiotic classes by poultry farmers [33].

The findings above point to inappropriate usage of antimicrobials in the study areas, which may have serious implications for public health. It is worth noting that any use of antibiotics is capable of selecting for antibiotic-resistant bacteria [23,34] but misuse and overuse are the major drivers [35]. In many LMICs, excessive misuse of antimicrobials is due to the easy availability of these drugs, which can be purchased without prescription [36]. Inappropriate use and misuse of antibiotics result in environmental contamination, which can lead to the introduction of ARGs and resistant bacteria into the human food chain and clinical environments [37]. Furthermore, the indiscriminate use of drugs and the failure to respect withdrawal periods may lead to the presence of residues in animal products. A recent study has reported high levels of sulphonamides in broiler meat (the mean residue level was 0.064 µg/g, which is above the maximum residual limit of 0.002 µg/g) [17,38]. The farmers in our study stated that they wanted more information about withdrawal periods and public health aspects of AMU (Table 5), indicating an opportunity for improvements.

The drug purchasing behavior of poultry farmers was described by the agrovet owners, such as listing clinical signs observed in flocks and seeking advice on products to use, presenting empty sachets, asking for drugs that cost about USD 1, and insisting on being sold specific veterinary products. These purchasing practices bring into focus the role that the interaction between the farmers and drug-sellers plays in the sale of antibiotics, and the need to consider it in interventions to reduce antimicrobial use. Improving farmers’ knowledge will not be sufficient; all actors must be included in the work to improve AMU and prevent negative public health effects. Indeed, a recent report has documented customers’ preferences as a driver for sale of antimicrobials in both human pharmacy and veterinary pharmacy retail outlets in Kenya [34]. A study in Uganda has also recently reported trade in antibiotics to be the one that contributes most to the profits of drug retailers [39]. There is an urgent need to build the knowledge capacity of agrovets regarding antimicrobial use and how this is linked to AMR, and the resulting effects on health (both human and animal) and ecosystems. Preventive measures, including advising farmers, should be prioritized, as these would prevent disease and reduce the need for antibiotic use.

The drug distribution chain involved farmers, agrovets, distributors, and pharmaceutical companies. The veterinary medicines supply chain was dominated by the agrovet shops that supplied drugs to farmers, and other animal health service providers comprising the private veterinarians and para-veterinarians. Veterinary outlets play a major role in the poultry product value chains in Kenya through provision of farm inputs and extension services [29,40]. Hence, it is important to include these in the work towards improved AMU. Indeed, the shops have been reported as major nodes for the supply of veterinary drugs in Tanzania [41]. These drug outlets are further supplied by local distributors. In Uganda, it is reported that retailers prefer farmers as their main customers because they pay higher prices as compared to animal health providers, who have better market information of the products from wholesalers [39]. The concept where veterinary products are pushed to the farmer has the potential to increase antibiotic consumption and eventually contribute to the development of AMR. Strengthening the role of veterinarians and other animal health providers is thus an important part of addressing this challenge, and this will be addressed in our future work building on this study.

The packages for drugs, including sachets, were disposed of through burning, burying, or by dumping in pit latrines by the farmers. Although some agrovets reported that they would always return the expired drugs to the distributors, other waste—including the sharps and packages—were disposed of in the local council dumping sites. Such inappropriate handling of waste may cause direct public health risks but may also contribute to environmental contamination and selective pressure for AMR in the environment.

This study provides baseline information for further work to improve AMU among the study participants. It also gives insights into the entire veterinary drug value chain and its key actors; information that is needed to design interventions for addressing overuse and misuse of antibiotics and other pharmaceuticals in animal production.

## 4. Materials and Methods

### 4.1. Study Sites

The study was implemented in Kajiado and Machakos counties, in the months of April and May 2021. The two counties border Nairobi County and are among the key suppliers of poultry meat that is consumed in the capital city, Nairobi. In each county, one sub-county was selected (considering the popularity of poultry production and proximity to Nairobi). In Kajiado, Kajiado North was selected and specifically these five wards, namely Ololua, Ngong, Olkeri, Nkaimurunya, and Ongata Rongai, all of which were included in the study. In Machakos, Machakos Central Sub-County was selected and seven wards, namely Machakos Town, Mumbuni, Mua, Mutituni, and Mavuti were selected. The locations of the study farms are presented in Figure 7. The total number of chickens owned by the study farms were 13,566 in Machakos and 96137 in Kajiado. The human population in the two study counties is 1,421,931 and 1,117,840 in Machakos and Kajiado, respectively. The population density is 235 per square kilometer for Machakos and 51 per km^2^ in Kajiado [42].

### 4.2. Study Design

This was a cross-sectional study involving questionnaire-based interviews with poultry farmers and key informant interviews with other value chain actors.

The interviews were performed by researchers from the International Livestock Research Institute and the University of Nairobi, in collaboration with representatives from the respective local government office. An official from the government office was included in the field team, who liaised with the farmers before the visit and decided which farms should be visited each day. Briefly, the veterinary department in each county was asked to provide a list of poultry farmers for the selected sub-counties. For Machakos, the list included 68 farmers (from which 50 were randomly selected for inclusion in the study). However, after interviewing 10 farms it was found that the list was biased, as it had names that had benefited from another poultry project where the farmers had received birds of the local “kienyeji” breed. An additional 15 farms were found through the help of the field contact affiliated to the livestock production department at the county. A snowballing approach was then instead used to attain the required numbers. Hence the list that was finally used was a combination of the farms initially provided, the additional ones from the field contact, and those from the snowballing. For Kajiado, all the farmers in the list provided (*n* = 50) were enrolled in the study. A sample size of 100 farmers was considered adequate, as the aim of the baseline survey was not to estimate any population parameter or assess the significance of any difference, but to obtain key data that would inform the design of the planned ICT intervention. Ethical considerations (gender aspects, power imbalance, personal integrity) were taken into account when designing the study and was part of the process in seeking the necessary study permits. All participants were ensured anonymity in the data analyses and publication and gave their informed consent before entering the study.

### 4.3. Interviews with Poultry Farmers

Interviews were based on a questionnaire designed for the study. To gain insights into the drug use by farmers, we compiled a list of frequently used drugs (including their pictures), which we updated during the field activity to include new products reported by farmers. Farmers were shown the pictures and asked a series of questions related to their use (whether they had seen or used the product, the last time the product was used, administration route, quantities given, etc.). A few questions related to the intervention were included and were meant to provide information to aid the design of the proposed ICT tools.

One adult person available on the farm at the time of the interviews, who was knowledgeable regarding flock management, was targeted for the interview. Written informed consent was sought from all participating farmers after introducing the research activity. The interviews lasted for about an hour. The questionnaire was in English, but the questions were translated into Swahili during the interviews.

### 4.4. Key Informant Interviews

Agrovet shops, animal health providers attached to these, and veterinarians working with the county governments, were considered in this component (*n* = 17). In addition, we interviewed two stakeholders who had previous experience of working in the pharmaceutical and feed industry sectors. For the agrovet outlets, a list was obtained from the county veterinary office. The number to enroll was based on their availability at the time of the interviews and willingness to participate in the study.

The interview guide included questions on the sources and use of veterinary drugs, the handling of expired products, the disposal of drug packages and containers, public health risks associated with antibiotic use, and their perceptions related to the proposed intervention.

### 4.5. Validation of Findings with Stakeholders

An online workshop was arranged to validate findings from the baseline activity and to solicit inputs to guide the design of the ICT intervention. The workshop was held on 15 August 2021 and lasted for two hours.

### 4.6. Data Analyses

The questionnaire data were downloaded as an MS Excel^®^ file and cleaned. The cleaning included checking the data for possible errors and preparing it for analysis. The data manipulations included generating new variables based on responses to specific elements of the questionnaire, for example: drug categories—dewormer (yes, no), antibiotic (yes, no), acaricide (yes, no), disinfectant (yes, no), antibiotic (yes, no); specific ingredients that the reported antibiotics was thought to contain—polymyxin (yes, no), tetracycline (yes, no), sulphonamides (yes, no), quinolone (yes, no), macrolide (yes, no); actual use of the products by the farmers (yes, no); drug application methods—water (yes, no), feed (yes, no), injection (yes, no); and the reasons why the drugs were used with yes or no options for each reason given. Antibiotic products were classified as either “single” or “multiple” depending on the ingredients the product contained. Statistical analyses were performed in Stata^®^ (version 17, StataCorp LLC, College Station, TX, USA) and were mainly descriptive (tabulations, frequency graphs). Thematic approaches were applied for the KII data.

## 5. Conclusions

Animal diseases and access to appropriate advice on how to manage them are among the challenges faced by smallholder farmers. Agrovets are an important source of inputs. An ICT system for communication about disease prevention and treatment, easy contact with animal health professionals, reporting of diseases, and drug use could support poultry production and poultry health while providing an opportunity for the monitoring of animal diseases and the use of veterinary drugs.

## Figures and Tables

**Figure 1 antibiotics-12-00905-f001:**
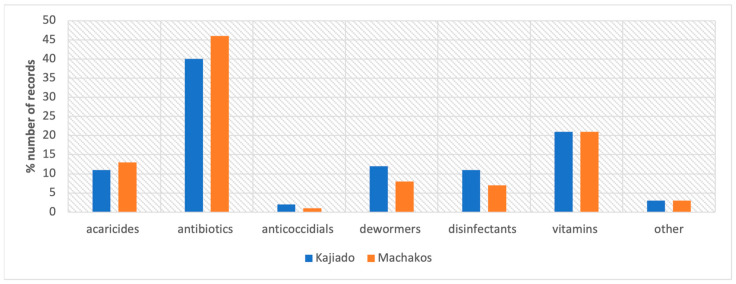
Drug use by 100 poultry farmers in Machakos and Kajiado counties, April–May 2021, percentages of all responses about used drugs in each county.

**Figure 2 antibiotics-12-00905-f002:**
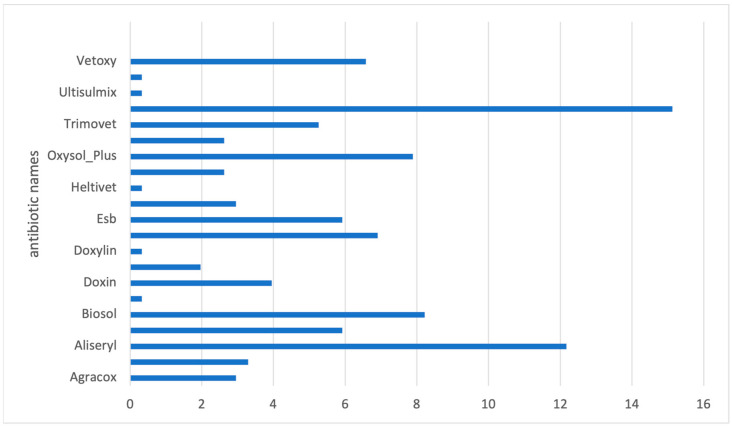
Antibiotic products (% of responses) reported in the study on 100 Kenyan poultry farmers, April–May 2021.

**Figure 3 antibiotics-12-00905-f003:**
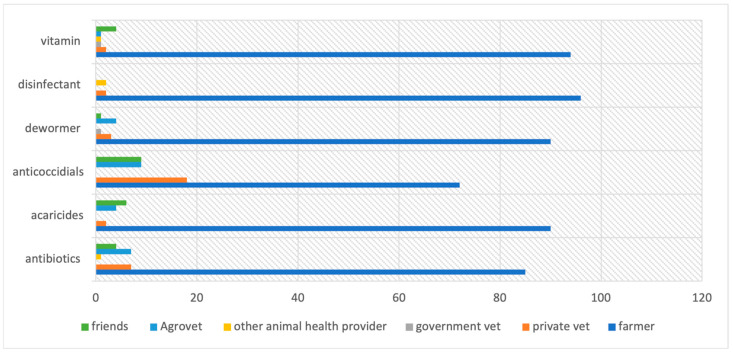
Administration of medical products, as reported by 100 Kenyan poultry farmers, April–May 2021.

**Figure 4 antibiotics-12-00905-f004:**
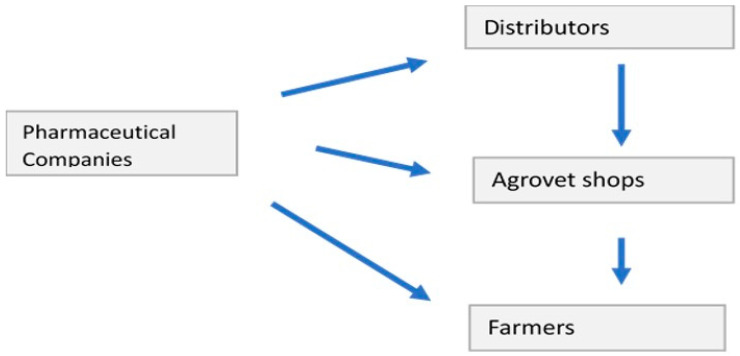
Distribution of poultry veterinary products in Machakos and Kajiado.

**Figure 5 antibiotics-12-00905-f005:**
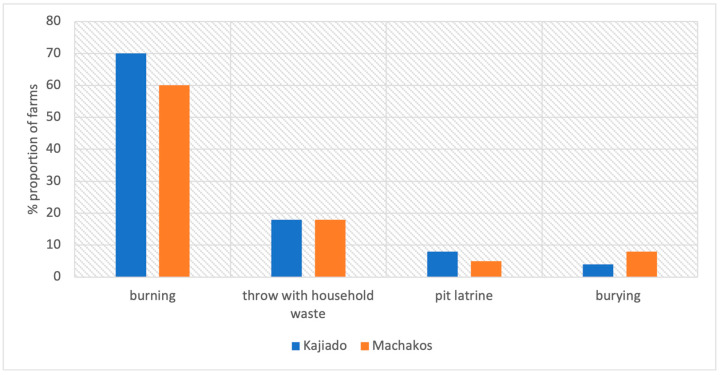
Disposal of empty drug containers and packages by 100 poultry farmers in Kajiado and Machakos Counties, April–May 2021.

**Figure 6 antibiotics-12-00905-f006:**
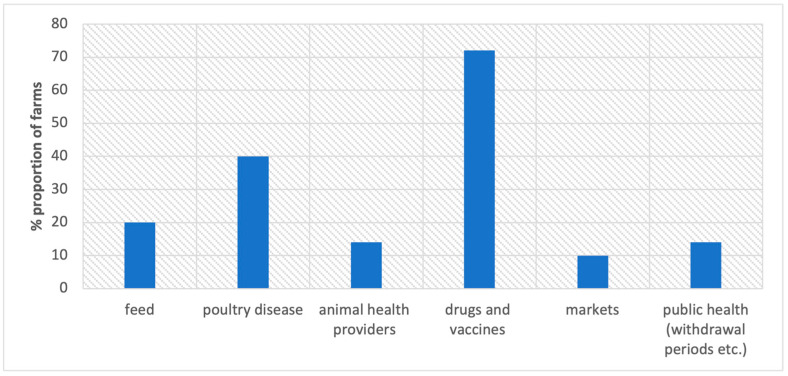
Information needs, as stated by the participating poultry farmers.

**Figure 7 antibiotics-12-00905-f007:**
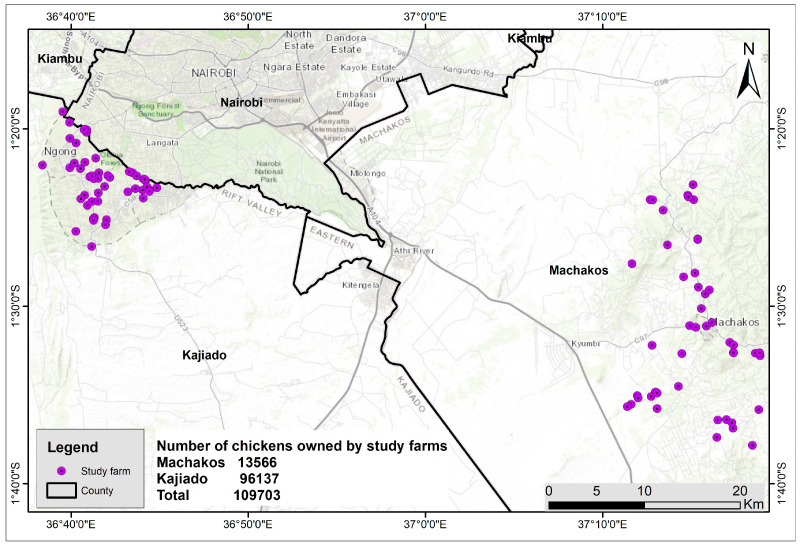
Map showing the location of study farms.

**Table 1 antibiotics-12-00905-t001:** Reasons for use of human drugs to treat animals, as reported by farmers using this practice.

Reason Given	Comments Given (If Any)
They work best	Eye infections, tetracycline
They work the same	Amoxycillin
It was recommended by a friend	For turkeys
Advised by a neighbor	Birds died, used only once
There is none for animals	For eye infection
Tried because other drugs were not working	For coccidiosis
If they fail to improve after using veterinary medicine	Especially respiratory disease, amoxycillin
Availability	Also works well
Because no eye ointment is available for chicken	For eye diseases

**Table 2 antibiotics-12-00905-t002:** Antibiotic products reportedly used by 100 Kenyan poultry farmers in a study April–May 2021 and their composition (single = one antibiotic class, multiple = >1 antibiotic class).

Trade Name	Multiple/Single Ingredient	Antibiotic Composition
Agracox	Multiple	Sulfadimerazine; sulfadiazine; pyrimethamine; furaltadone; vitamins
Doxin	Multiple	Doxycycline; tylosin
Doxy-tyl	Multiple	Doxycycline; tylosin
Doxylin	Multiple	Doxycycline; tylosin
Neoxy vitamin	Multiple	Oxytetracycline; neomycin; vitamins
Aliseryl	Multiple	Oxytetracycline; streptomycin; erythromycin; colistin; vitamins
Oxysol Plus	Single	Oxytetracycline; vitamins
Trimovet	Multiple	Sulphamethoxazole; trimethoprim
Tetracolivit	Multiple	Oxytetracycline; colistin; vitamins; calcium
Agraryl	Multiple	Oxytetracycline; streptomycin; erythromycin; colistin; vitamins
Egocin	Single	Oxytetracycline; vitamins
Biosol	Multiple	Trimethoprim; sulphamethoxazole
Esb	Single	Sulfaclozine sodium monohydrate
Tylodoxy	Multiple	Doxycycline; tylosin
Veta Oxy	Single	Oxytetracycline
Vetoxy	Single	Oxytetracycline
Ultisulmix	Multiple	Trimethoprim; sulphamethoxazole
Fosbac	Multiple	Fosfomycin; tylosin
Diaziprim	Multiple	Sulfadiazine; trimethoprim
VetTrim	Multiple	Sulphamethoxazole; trimethoprim

**Table 3 antibiotics-12-00905-t003:** Drug administration in 100 smallholder poultry farms in Machakos and Kajiado counties, April–May 2021. Figures represent % of responses by farmers.

Drug Administrator	Kajiado (*n* = 371)	Machakos (*n* = 335)
Self (by farmers themselves)	85	95
Animal health providers (government, private)	9	2
Agrovets	3	7
Friends	6	0.2

**Table 4 antibiotics-12-00905-t004:** Reasons for use of the drugs and treatment options considered by the 100 Kenyan poultry farmers, April–May 2021, *n* = 706.

Reason for Medication	Responses (%)	Treatment Given (*n*) *
Detoxification	10 (1.4)	Vitamins (10)
Boost growth	112 (16)	Antibiotics (27), dewormers (2), disinfectant (1), traditional (1), vitamins and other products (78)
Infectious coryza	10 (1.4)	Antibiotics (8)
Diarrhea	82 (11.6)	Antibiotics (75), anticoccidials (5)
Eye infection	9 (1.3)	Antibiotics (8)
Coccidiosis	15 (2)	Antibiotics (13), anticoccidials (2)
Ectoparasites	63 (8.9)	Acaricides (63)
Respiratory problems	53 (7.5)	Antibiotics (49), dewormers (2), vitamins and other products (2)
Helminths	40 (5.6)	Antibiotics (1), dewormers (37), vitamins and other products (1)
Dropping feathers/general body weakness	56 (7.9)	Antibiotics (48), dewormers (1), vitamins and other products (6)
Stress	22 (3.1)	Acaricides (1), antibiotics (10), anticoccidials (1), vitamins and other products (9)

* Excludes options given as “other”.

**Table 5 antibiotics-12-00905-t005:** Disposal of empty drug packages and containers, according to 17 key informants.

Route of disposal	Description
Pit latrine	Used especially for needles and glass bottles.
County waste disposal system	Public dumping sites where the waste is handled by the city council (same as with other waste). The municipal council may collect the waste from the agrovet and handle it in the same way as other garbage. Empty bottles may also be put in trash bags and given to the county, separate from other waste. Before disposing, the agrovet keeps the waste in a carton that is kept closed.
Burn or bury	Applies to plastic containers, packages, and syringes. Needles are put in water bottles and taken home for disposal.
Incineration	When there is an arrangement, institutions offering incineration services may visit the agrovet and collect needles for disposal. A charge of KES 60 (USD 0.46)/kg was reported.

**Table 6 antibiotics-12-00905-t006:** Detailed description of the Kenyan poultry farmers’ information needs.

Information Category	Description
Feed	Feeding formula and formulation (how to formulate their own feeds and how to ration the feeds to maximize growth and profit).
Poultry diseases	Symptoms of common diseases and how best to prevent and manage them. Notification in case of outbreaks. If there are seasonal diseases in the area, information on when they usually occur and how best to manage the diseases.
Animal health providers	The nearest animal health providers and their contact addresses. A list of genuine agrovets where farmers can be assisted.
Drugs and vaccines	Vaccination schedule, what vaccine to give and when, and where to access them. Available drugs that can be used to manage common diseases in their localities. Specific information on each drug, such as prescription and withdrawal period.
Markets	Availability of markets for their poultry and poultry products and/or advice on how to market them.
Public health	Withdrawal period of drugs. How to prevent infections without use of antibiotics. Side effects of using certain drugs. How best to store veterinary products.
Farmer groups	A platform where farmers can connect and share their experiences and educate each other.

## Data Availability

Data may be available on reasonable request from the corresponding author.

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
