# Peer review of "Antimicrobial Use by Peri-Urban Poultry Smallholders of Kajiado and Machakos Counties in Kenya"

_antibiotics, 2023, doi:10.3390/antibiotics12050905_

Round 1

Reviewer 1 Report

The abstract and introduction parts need to improve in a more significant way.

What about ethical concerns?

Statistical analysis is needed to describe

The figure quality is poor and can be improved.

Author Response

Thank you for these comments. The first one is not so specific, but we have improved the abstract and introduction based on the comments from the other reviewers, and provided additional clarification. We hope this will be what was sought by this reviewer.

As stated in the Institutional Review Board Statement: the ethical policies of the journal have been adhered to and approval was obtained from the Institutional Research Ethics Committee. We have also explained that informed consent was obtained from all participants. A clarification of this has been added at the end of section 4.2, lines 518-521

The third comment has been addressed by clarifying how the statistical tests were done (lines 579-581). However, we note that the objective was not to make any statistical comparisons hence no detailed analyses were done.

The figures have been re-formatted to improve resolution.

Reviewer 2 Report

I congratulate the authors for the relevance of the topic - the use of antibiotics in smallscale poultry production, but I believe that for publication in Antibiotics it should have greater scientific consistency.

The use of surveys in two regions, not being known the universe of producers nor their characterization, the sample selection (with the deviation mentioned), the absence of a simple statistical treatment (gender, age, education), the absence of reference to implications or knowledge about antibiotic resistance and its impact on public health, reduces the scientific relevance of the article.

Therefore, I consider it relevant a profound restructuring, a deepening of data, its extension to knowledge about resistance and its impact on Public Health, in order to be publishable.

Author Response

We thank the reviewer, especially for pointing out this omission and have added more background information about the two study regions (although there is not much publicly available data on the poultry production) in section 4.1 (lines 504-507). In reference to the comment on the lack of simple statistical treatment, we note that the aim of the survey was not to statistically assess any difference, but to get key data that would inform design of the planned ICT intervention.

As our further work will aim to improve the AMU practices in the study region, it was important to collect baseline data on these practices. As seen from the references in the discussion, our results appear to be representative of the practices in and around the study region. This has been further clarified in the end of the Discussion.

Reviewer 3 Report

All my comments are in the attached file. 

BR

Author Response

Thank you. We have edited the manuscript according to the suggested changes in most parts, except for:

Table 2. the vitamin content was not specified and hence this information cannot be provided

Figure 6. This was not information collected from the poultry farmers, it describes the information they said they needed. The figure legend has been clarified as it apparently could be misunderstood.

Round 2

Reviewer 2 Report

I thank the authors for their responses to my comments and the introduction of some suggestions that have enhanced the article.

In my opinion, the weakness remains the reduced scientific validity, given the absence of a statistical treatment, even if it is a survey or if the methodological description has now been improved.

The fundamental scientific impact of works of this type is thus lost and we have another interesting description of antibiotic use practices in a certain population.